# Study of the Effect of NaOH Treatment on the Properties of GF/VER Composites Using AE Technique

**DOI:** 10.3390/ma17061407

**Published:** 2024-03-19

**Authors:** Lin Ming, Haonan He, Xin Li, Wei Tian, Chengyan Zhu

**Affiliations:** Key Laboratory of Advanced Textile Materials and Preparation Technology of the Ministry of Education, College of Textiles Science and Engineering, Zhejiang Sci-Tech University, Xiasha Campus, Hangzhou 310018, China; mlms@foxmail.com (L.M.); hehaonan1998@foxmail.com (H.H.); lixin_starry@foxmail.com (X.L.); tianwei_zstu@126.com (W.T.)

**Keywords:** acoustic emission technology, glass fiber, interface modification, mechanical properties, damage classification

## Abstract

The purpose of this study is to use acoustic emission (AE) technology to explore the changes in the interface and mechanical properties of GF/VER composite materials after being treated with NaOH and to analyze the optimal modification conditions and damage propagation process. The results showed that the GF surface became rougher, and the number of reactive groups increased after treating the GF with a NaOH solution. This treatment enhanced the interfacial adhesion between the GF and VER, which increased the interfacial shear strength by 25.31% for monofilament draw specimens and 27.48% for fiber bundle draw specimens compared to those before the GF was modified. When the modification conditions were a NaOH solution concentration of 2 mol/L and a treatment time of 48 h, the flexural strength of the GF/VER composites reached a peak value of 346.72 MPa, which was enhanced by 20.96% compared with before the GF was modified. The process of damage fracture can be classified into six types: matrix cracking, interface debonding, fiber pullout, fiber relaxation, matrix delamination, and fiber breakage, and the frequency ranges of these failure mechanisms are 0~100 kHz, 100~250 kHz, 250~380 kHz, 380~450 kHz, 450~600 kHz, and 600 kHz and above, respectively. This paper elucidates the fracture process of GF/VER composites in three-point bending. It establishes the relationship between the AE signal and the interfacial and force properties of GF/VER composites, realizing the classification of the damage process and characterizing the mechanism. The frequency ranges of damage types and failure mechanisms found in this study offer important guidance for the design and improvement of composite materials. These results are of great significance for enhancing the interfacial properties of composites, assessing the damage and fracture behaviors, and implementing health monitoring.

## 1. Introduction

Glass fiber-reinforced vinyl ester resin composites (GF/VER) are a type of fiber-reinforced resin matrix composite that fall under the category of thermoset polymer matrix composites [1]. These composites possess excellent properties, such as high specific strength, corrosion resistance, high-temperature resistance, and good plasticity. They have wide applications in aerospace, wind power generation, infrastructure, and other industries [2,3,4]. However, the smooth surface of GF, their chemical inertness, and low wettability contribute to weak bonding at the interface between the fibers and the resin. As a result, the fibers tend to slip out of the resin when the GF/VER composite material is subjected to stress [5,6]. The interface between fibers and resins affects the load transfer capability from fiber-reinforced materials to polymer resins and the mechanical properties of composite materials [7,8,9]. The interface between fibers and resins affects the load transfer capability from fiber-reinforced materials to polymer resins and the mechanical properties of composite materials [10,11,12]. Therefore, a surface modification of the GF that enhances the interface and mechanical properties of GF/VER composite materials holds significant potential for application.

At present, the methods of fiber surface modification mainly include coupling agent modification [13], surface coating modification [14], plasma modification [15], and acid and alkali modification [16]. Among these methods, coupling agents, surface coatings, and plasma modification have less impact on the fiber, but their modification effects are temporary and can degrade over time. Additionally, these methods require more complex operations and incur higher costs [17,18,19]. Compared to the above-mentioned modification methods, acid and alkali modification have several advantages. They are easy to operate, cost-effective, and produce a noticeable modification effect on the fiber surface. However, a disadvantage of this method is that it can weaken the strength of the fiber [20]. Conventional inspection methods are unable to accurately detect the entire process of damage development in composites in real-time. Research and engineering practice has shown that the use of non-destructive testing (NDT) techniques for assessing damage in composites is more appropriate. The NDT techniques include ultrasonic testing [21], X-ray testing [22], and AE testing [23]. These sensors have high sensitivity, fast response, and high accuracy. The main advantage of AE technology, compared to the other two techniques, is its ability to detect the time, location, type, and severity of material damage [24].

Fiber-reinforced composites exhibit anisotropic and non-homogeneous properties, which complicates the analysis of acquired AE signals due to the complex propagation of waveforms in the presence of damage [25,26]. Hence, the study of damage mechanisms and damage extension processes in fiber-reinforced composites poses greater challenges compared to isotropic materials [27,28,29]. The selection of an appropriate signal processing method is of utmost importance in the detection, analysis, and study of acoustic emission (AE) signals, as it significantly contributes to enhancing the accuracy of detection. Examples of these methods include fast Fourier transform (FFT) [30], wavelet analysis [31], and neural networks [32]. The FFT technique has demonstrated its ability to effectively classify the type of material damage [33,34,35]. Wavelet analysis proves to be highly effective in the detection of wear characteristics in structures [36,37]. Neural networks have been found to exhibit superior accuracy in the localization of AE [38].

Over the years, several researchers have employed AE techniques to examine the alterations in damage and failure mechanisms caused by structural modifications in composites [39,40,41]. For instance, Anaya-Ramirez et al. [42] conducted a study on the microscopic damage mechanisms of HDPE/surface-modified Henequen fiber-reinforced composites. They utilized the AE technique to analyze the characteristics of stress wave signals in composite specimens and evaluate the adhesion level at the interface. Scholey et al. [43] quantitatively measured the amplitude and angle changes of AE signals caused by matrix cracking and delamination in composite panel specimens. Bussiba et al. [44] investigated the micro-damage and failure mechanisms of three distinct composite materials using AE technology. Sasikumar et al. [45] studied tensile specimens of unidirectional carbon/epoxy composites using the AE technique to analyze the various damage modes of the composites. Nevertheless, previous studies have primarily concentrated on investigating the damage mechanism of materials and the analysis of AE signals. There is a scarcity of research that has established a correlation between the modification effect and AE signals.

Based on the above studies, this paper used a NaOH solution to perform a surface modification on the GF. The interfacial properties of the GF and VER were examined by the single fiber pullout and fiber bundle pullout tests. Subsequently, the GF/VER composites were prepared with unmodified and modified GF as reinforcement, and the flexural strength of the specimens was tested. The microscopic examination of the loaded specimens’ damage and fracture surfaces was conducted using a scanning electron microscope. Additionally, AE techniques were employed to collect the signals generated during the damage process. These signals were then analyzed using various methods, such as AE characteristic signals, K-means cluster analysis, and FFT. The relationship between the AE signal and the interfacial and mechanical properties of the GF/VER composites has been established. The changes in the interfacial and mechanical properties of the GF/VER composites after the GF was modified were analyzed. Damage processes were classified and characterized mechanistically. The results could be used for the interfacial modification of GF/VER composites, as well as for damage fracture characterization and health monitoring under bending loads.

## 2. Materials and Methods

### 2.1. Materials and Equipment

The materials and equipment used in the experiment are listed in Table 1:

### 2.2. Preparation of Modified Specimens

The GF fabrics were soaked in acetone solution for 24 h to remove impurities from the fiber surface. Afterward, the fabrics were washed with ethanol solution and distilled water and then dried in a vacuum drying oven. The dried GF fabrics were subjected to immersion in NaOH solutions with concentrations of 1, 2, and 3 mol/L for durations of 24, 48, 72, and 96 h. Subsequently, the fabrics underwent washing with ethanol solution and distilled water, followed by a final drying process.

### 2.3. Fiber Tensile Property Test

The breaking strength of modified and unmodified GF monofilaments was tested using an electronic single-fiber tester with a specimen spacing of 20 mm and a loading speed of 5 mm/min. The breaking strength of the monofilaments was calculated using the following formula [46]:$$\sigma_{1}=\frac{F_{1}}{\rho_{1}}$$
where σ_1_ [cN/dtex] is the breaking strength of a single fiber, F_1_ [cN] is the load of single fiber breaking, and ρ_1_ [dtex] is the linear density of a single fiber. The average value was calculated by testing the breaking strength of 50 single fibers for each group of specimens.

The breaking strength of modified and unmodified GF fiber bundles was tested using a universal material testing machine with a specimen spacing of 250 mm and a loading speed of 200 mm/min. The breaking strength of the fiber bundles was calculated using the following formula:$$\sigma_{2}=\frac{F_{2}}{\rho_{2}}$$
where σ_2_ [N/tex] is the breaking strength of fiber bundles, F_2_ [N] is the load of fiber bundles breaking, and ρ_2_ [tex] is the linear density of fiber bundles. The average value was calculated by testing the breaking strength of 10 fiber bundles for each group of specimens.

### 2.4. Fiber Pull-Out Test

#### 2.4.1. Preparation of Fiber Pull-Out Sample

The specimens for single fiber pullout and fiber bundle pullout are depicted in Figure 1a,b, and the corresponding preparation process is outlined as follows: Two PVC foam boards were selected to prepare the fiber pullout specimen mold. Small holes were evenly spaced on the boards to create the monofilament pullout specimen (small circular hole diameter of 0.5 mm) and the fiber bundle pullout specimen (small circular hole diameter of 1.5 mm). The GF was positioned carefully between two wooden boards, and the VER was injected fully into the holes with a syringe. After 24 h had elapsed, the resin had fully cured. The specimens were extracted from the mold.

#### 2.4.2. Interfacial Shear Strength Test

The interfacial shear strength of the monofilament tensile specimens was measured by an electronic single-fiber strength tester, while the interfacial shear strength of the fiber bundle tensile specimens was tested by a universal material testing machine. The upper collet of the instrument is loaded with fiber, and the lower collet is loaded with resin. The upper collet is gradually raised until the fibers are pulled out from the resin, at which point the pullout force is measured. The loading speed was 1 mm/minute, and the shear strength was calculated using the following formula [47].
$$\tau =\frac{F}{\pi dL}$$
where *τ* [MPa] is the shear strength, *F* [N] is the load of fiber pulling out from the resin matrix, *d* [mm] is the fiber diameter, and *L* [mm] is the embedded length, taking the average of five samples.

### 2.5. Three-Point Flexural Test

#### 2.5.1. Sample Preparation and Testing

The GF/VER composite specimens are depicted in Figure 2, and their preparation process is outlined as follows: The GF/VER composites were prepared by vacuum-assisted resin infusion (VARI) molding technique by combining GF fabric with VER under standard conditions (temperature of 25 °C and humidity of 65%). The mass ratio of vinyl ester resin to hardener is 100:2, and it is made into composite laminates with a thickness of 2 ± 0.1 mm after curing for 24 h at room temperature.

Samples of 60 × 15 × 2 mm were cut out of the middle of the GF/VER composite using a cutting machine. The samples were placed on a universal testing machine to test the three-point bending performance. The span was set to 32 mm. The lower indenter was held stationary, and the upper indenter was loaded upward at a rate of 5 mm/min. The destructive force resulting from the fracture of the samples at the middle position was measured.

#### 2.5.2. Bending Strength Calculation

The flexural strength σ*_f_* is the maximum bending stress experienced by a specimen at the point of failure or maximum load during a bending test. When subjected to bending loads, composite materials usually experience damage in the form of matrix cracking and delamination. This suggests that the bond strength between the matrix and the interface plays a crucial role in determining the flexural performance of the composite material. Therefore, the magnitude of the flexural strength can reflect the interface and mechanical properties of the composite material. The calculation formula is as follows [48]:$$\sigma_{f}=\frac{3pl}{2b\cdot h^{2}}$$
where σ*_f_* [MPa] is the bending strength, *p* [N] is the failure load, *l* [mm] is the span, *h* [mm] is the thickness of the specimen, and *b* [mm] is the width of the specimen, taking the average of five samples.

### 2.6. SEM Imaging

The morphology of fracture surfaces of GF/VER composites under different modification conditions was observed using scanning electron microscopy after being subjected to bending loads.

### 2.7. AE Signal Acquisition

#### 2.7.1. AE Data Acquisition Device

The AE acquisition system consists of an Express-4 AE transmitter, and 2/4/6 preamplifiers, and utilizes two channels. The analog filter has a lower limit of 100 kHz and an upper limit of 3 MHz. The waveform settings are 1 MPS, 256, and 1 kHz. The specific parameter settings are shown in Table 2.

#### 2.7.2. Acquisition of AE Signals of Specimen Damage

Figure 3 shows a schematic diagram of the AE signal when a GF/VER composite material is subjected to a three-point bending load. Referring to the standard ASTM D790-02 [49], the GF/VER composites were cut into specimens with dimensions of 15 mm × 80 mm × 2 mm. The bending properties of the specimens were tested by a universal material testing machine with a set span of 32 mm and a loading speed of 5 mm/min. During the testing process, two AE sensors were installed on the surface of the specimen to accurately receive and record the propagating AE signals. The surface between the transducer and the sample was filled with petroleum jelly to serve as a coupling agent, enhancing the transmission rate of the AE waveform and achieving effective acoustic coupling. The tape was used to secure the transducer to the surface of the specimen, effectively minimizing the impact of noise. A lead break test was performed to determine the sensitivity of the device, and then the ambient noise was tested to establish the corresponding threshold value.

#### 2.7.3. Fiber-Reinforced Composites Damage Process

As shown in Figure 4, the main sources of AE signals in fiber-reinforced composite materials include matrix cracking, interface debonding, fiber pullout, fiber relaxation, fiber breakage, and matrix delamination. During the entire damage process of composite materials, damage signals may be generated from a single source or multiple sources simultaneously [50,51,52].

### 2.8. AE Signal Analysis Method

#### 2.8.1. b-Value

The b-value holds significant importance as a parameter in AE experiments, as it quantifies the proportion of small AE events to large events that occur during the rupture process of composite materials [53]. The change in b-value can reflect the damage process of composite materials and is one of the important indicators of damage. The calculation formula is as follows [54]:$$lgN=a-b(\frac{A_{dB}}{20})$$
$$b=\frac{\sum_{i=1}^{m}M_{i}\sum_{i=1}^{m}\mathrm{lg}N_{i}-m\sum_{i=1}^{m}\mathrm{lg}N_{i}}{m\sum_{i=1}^{m}M_{i}^{2}-{\left(\sum_{i=1}^{m}M_{i}\right)}^{2}}$$
where *A_dB_* is the AE event’s peak amplitude [dB], *N* is the incremental frequency, and *a* and *b* are empirical constants estimated by linear curve fitting.

#### 2.8.2. K-Means

The K-means clustering algorithm is an unsupervised learning method with many applications. It divides the dataset into K different clusters. The k data points are randomly selected as the initial centroids, and then each data point is assigned to the cluster that is closest to its centroid. Subsequently, the centroid of each cluster is recalculated based on the average of the values of all the points in that cluster. This procedure is iterated until either convergence is achieved or the maximum number of iterations is reached [55,56,57].

#### 2.8.3. Fast Fourier Transform

The AE signal exhibits transience and randomness, placing it in the category of stationary random signals. Usually, there are two methods of analysis: time domain and frequency domain. The time domain analyzes the instantaneous amplitude of a signal over time, while the frequency domain examines the relationship between the amplitude and frequency of each frequency component of the signal. Time-domain descriptions cannot effectively capture the characteristics of signals because transient and random signals are not only time-dependent but also frequency and phase-dependent, among other factors.

Therefore, the frequency structure of the signal needs to be further analyzed and described in the frequency domain. The two analysis methods can be converted to each other through the fast Fourier transform. The FFT of a time domain signal is [58]:$$Y(f)=\int_{-\infty }^{\infty }y(t)e^{-j2\pi ft}dt$$
where Y(f) is the frequency domain of the signal, y(t) is the time domain of the signal, and f is the frequency.

As depicted in Figure 5, researchers and scholars have classified damage types based on the peak frequencies in the AE signals. Each frequency range corresponds to a specific damage type, and different classification results may arise due to variations in specimen materials and classification methods.

## 3. Results and Discussion

### 3.1. Tensile Performance Test Results

Figure 6 shows the changes in the breaking strength of unmodified and modified GF. The analysis indicates that following treatment with a sodium hydroxide solution, GF monofilaments and fiber bundles exhibit a similar decreasing trend in breaking strength. When the concentration of sodium hydroxide is low and the treatment time is short, the change in GF breaking strength is not significant. As the concentration of sodium hydroxide solution increases and the treatment time lengthens, the breaking strength of the GF gradually decreases. It can be concluded that the reaction between the GF and NaOH will weaken the strength of the fiber. The damage will be more severe with a higher concentration and longer treatment time.

### 3.2. Interfacial Shear Strength Test Results

The impact of sodium hydroxide solution treatment on the interfacial properties of the composites is illustrated in Figure 7. As the treatment time increased, the interfacial shear strength of the fibers gradually increased, and then decreased after reaching a certain value. In the alkali solution with a concentration of 2 mol/L and a treatment time of 48 h, the interfacial shear strength of monofilaments and fiber bundles reached the maximum values of 5.00 MPa and 29.04 MPa, respectively. The analysis shows that the interfacial properties between the modified GF and the resin have been significantly improved. This improvement is mainly manifested in the enhancement of the bonding force between the GF and resin, which makes it difficult for the GF to be pulled out or fall off from the resin matrix when subjected to tensile force. The good interfacial bonding between the modified GF and the resin helps to achieve more effective energy dissipation during the stress transfer process, thereby reducing the probability of GF pullout in the resin matrix.

### 3.3. Three-Point Bending Strength Test Results

The bending load variation of the modified and unmodified specimens is shown in Figure 8. The analysis shows that the flexural strength of GF/VER composites increases gradually with the increase in NaOH solution concentration and the prolongation of treatment time. However, it then decreases gradually after reaching a certain critical point. Combined with Figure 7, it can be observed that when the concentration of the NaOH solution is low and the treatment time is short, the fiber-resin interface is not sufficiently tight. This results in the GF slipping out of the resin during the damage process of the composite material. When the concentration of the NaOH solution is 2 mol/L and the treatment time is 48 h, the bonding effect between the fiber and the resin is optimal. When the sample is loaded, the fiber cannot slip out of the resin easily. Additionally, the flexural strength of the specimen reached a peak value of 346.72 MPa, which represented a 20.96% improvement compared to the pre-modification state. When the concentration of NaOH is excessively high and the treatment time is excessively long, the flexural strength of the composites subsequently decreases because the mechanical properties of the GF are damaged after being treated with NaOH. This occurs because when NaOH reacts with the silicon–oxygen bonds on the surface of GF, it produces sodium silicate and water, along with a significant amount of heat. The heat accelerates the movement of molecules inside the GF, breaking the original silicon–oxygen bonds and causing a reduction in the mechanical properties of the GF [60]. GF is the reinforcement in composite materials and plays a crucial role in determining the overall mechanical properties of the material. Therefore, when the GF is damaged, it will also impact the overall mechanical properties of the GF/VER composite material. In conclusion, it can be inferred that when GF interacts with NaOH, it enhances the roughness of the GF surface and increases the interface strength between the GF and VER. However, the damage to GF will also affect the overall mechanical properties of the GF/VER composite material.

### 3.4. Fracture Surface Morphology of GF/VER Composites

An electron micrograph of the fracture surface of the specimen after being damaged by a three-point bending load is shown in Figure 9. When a specimen is subjected to a three-point bending load, various types of damage occur simultaneously. Different colored markers in the figure represent various damage modes: purple indicates matrix cracking, yellow indicates interface debonding, green indicates fiber pull-out, blue indicates fiber relaxation, orange indicates fiber breakage, and red indicates matrix delamination. Examination of the electron microscope Figure 9a reveals that the surface of the GF in the fracture section of the unmodified GF/VER composite is relatively smooth. This indicates that the specimen is vulnerable to fiber pullout damage and has very little resin adhering to the GF surface, suggesting weak interfacial adhesion of the unmodified GF/VER composite. After modifying the GF, the electron microscope images shown in Figure 9b–d indicate that increasing the concentration of NaOH solution results in a rougher fiber surface, while the electron microscope images in Figure 9b–k show that prolonging the treatment time produces a similar effect. Furthermore, as the fibers are withdrawn, more resin adheres to the fiber surface, indicating enhanced adhesion between the resin and the matrix. The enhanced adhesion serves to inhibit the easy sliding of fibers and matrix delamination. However, as shown in Figure 9m, the high concentration of NaOH solution and prolonged immersion will weaken the strength of the glass fibers and lead to fiber breakage, ultimately affecting the overall performance of the composite. The results from Figure 9a–m indicate that the interfacial bonding of GF/VER composites is optimal when the concentration of the NaOH solution reaches 2 mol/L and the treatment time reaches 48 h. These results indicate that the rough surface of the modified GF facilitates the mechanical bonding between GF and VER.

### 3.5. Principle of Reaction between NaOH and GF

The alkali resistance of the E alkali-free GF is comparatively lower than that of the medium-alkali and alkali-resistant GF. It is also more prone to react with the NaOH solution. Therefore, the GF utilized in this study is E alkali-free GF. The main structure of the GF is primarily composed of SiO_2_, with CaO, Al_2_O_3_, and B_2_O_3_ also present in significant amounts. The process of the reaction between this type of GF and NaOH is as follows:(1)$$\left(B_{2}O_{3},{Al}_{2}O_{3},{SiO}_{2}\right)CaO+H_{2}O\to \left(B_{2}O_{3},{Al}_{2}O_{3},{SiO}_{2}\right)+{Ca}^{2+}+2{OH}^{-}\;(\mathrm{GF})\;\;\;\;\;\;\;\;(\mathrm{reactive})\;$$
(2)$$\begin{array}{l} \left(B_{2}O_{3},{Al}_{2}O_{3},{SiO}_{2}\right)+2{OH}^{-}\to {\left[{\left(B_{2}O_{3},{Al}_{2}O_{3},{SiO}_{2}\right)\left(OH\right)}_{2}\right]}^{2-}\\ \;(\mathrm{reactive}) \end{array}$$
(3)$$\begin{array}{r} {\left[{\left(B_{2}O_{3},{Al}_{2}O_{3},{SiO}_{2}\right)\left(OH\right)}_{2}\right]}^{2-}+{Ca}^{2+}\to Ca{\left(B_{2}O_{3},{Al}_{2}O_{3},{SiO}_{2}\right)\left(OH\right)}_{2}↓\\ (\mathrm{gel})\;\; \end{array}$$

Reaction Equation (1) describes the process of diffusion and hydrolysis of calcium ions occurring outside the GF network. The reaction (2) involves the main components SiO_2_, Al_2_O_3_, and B_2_O_3_ in the GF reacting with hydroxide ions, resulting in the formation of [(B_2_O_3_, Al_2_O_3_, SiO_2_) (OH)_2_]^2−^. The reaction (3) involves the reaction between [(B_2_O_3_, Al_2_O_3_, SiO_2_) (OH)_2_]^2−^ and diffusing calcium ions, resulting in the formation of a gel with low solubility. The reaction will continue in the forward direction until all the calcium ions are completely consumed. Combining the scanning electron microscope photos of the damaged fracture surface of the GF/VER composite in Figure 9, it can be inferred that the protrusions formed on the surface of the GF after reacting with NaOH are the Ca(B_2_O_3_, Al_2_O_3_, SiO_2_)(OH)_2_ gel. Figure 10 illustrates the binding force between the modified GF and VER. The protrusions on the fiber surface create a riveting connection with the resin. The hydroxyl groups on the surface of the GF react with the epoxy groups in the VER, resulting in chemical bonding [61]. During this chemical reaction, the epoxy group exhibits high reactivity and can undergo a reaction with the hydroxyl group. The reaction results in the formation of a chemical bond that exhibits relatively strong and stable characteristics. This chemical bond is commonly referred to as a hydrogen bond, which facilitates a strong connection between the GF and the VER [62,63,64]. When the GF is modified and made into composites, a series of complex molecular entanglements between the GF and VER occur during the curing process of the composites, and the reaction primarily occurs at the interface, which makes it difficult to analyze qualitatively and quantitatively by intuitive methods, so this paper uses AE technology to analyze the bonding effect between the GF and VER in a more accurate and detailed way.

### 3.6. Characterization of AE Signals under Three-Point Bending Loads

The load-time curves versus the AE characterization parameters are shown in Figure 11. There is a strong correlation between the load-time curves and the AE signals corresponding to all phases of damage onset, evolution, and extension in GF/VER composites subjected to three-point bending loads. When the GF/VER composite is loaded, the transducer receives an increased AE signal with the load. It can be observed that when the bending load on the composite material reaches its maximum value, the energy, count, and b-value also reach their maximum values. This indicates that significant damage occurs in the composite material at the point of highest load, resulting in the release of a large number of AE signals.

Generally, the three-point bending load process can be divided into three stages: the stable growth period, the failure period, and the unloading period. These stages correspond to stages I, II, and III in the figure, respectively. The b-value fluctuates up and down as the load increases. Stage I is the initial stage of the three-point bending load application, during which the overall load on the composite material increases gradually. Small-scale microcracks dominate the material surface, and the overall b-value remains relatively stable with a sparse curve. In stage II, as the three-point bending load increases, the microcracks on the surface of the composite material merge to form larger cracks continuously. AE sources are clustered along the fracture plane, alternating between large and small events, with a higher proportion of large events. The b-value fluctuates dramatically within a certain range, overall decreasing, and the curve of the b-value is denser. In stage III, the final stage before failure occurs, the b-value decreases sharply and the b-value curve becomes sparse. This stage is characterized primarily by significant fracture events. As the concentration of the NaOH solution increases and the treatment time grows, the fluctuation of the b-value in Stage II becomes more pronounced, and the curve becomes more intense. This phenomenon indicates that the interfacial bonding strength of the modified GF/VER composites increases, which makes the magnitude events alternate more frequently during the loading process of the composites.

As depicted in Figure 11a, the unmodified specimen exhibits a significant number of energy signals and counting signals during the loading phase, suggesting a substantial level of damage to the specimen. The significant fluctuations in the b-value during stage II indicate that there are more frequent alternations between major and minor injury events. The decreasing trend of the b-value in stage III is very obvious, indicating that the failure stage of the specimen occurs rapidly. From Figure 11g, it can be seen that the energy counting signal of the specimen is lower during the stable growth period. However, the energy and counting signal increase significantly when the specimen enters the destructive period, and the energy corresponding to the highest point of the load also increases substantially. This indicates that the damage situation of the modified GF/VER composite material has improved. When damage occurs, it releases more energy. This is due to the increase in the concentration of NaOH solution and the extension of processing time, which alters the roughness of the GF surface and enhances the bonding strength between the GF and VER. From Figure 11m, it can be seen that higher concentrations of NaOH and longer processing times result in greater damage to the GF surface. The signals of matrix cracking, fiber breakage, and matrix delamination emitted by GF/VER increase accordingly when subjected to a load. The sharp fluctuations in the b-value during stage II and the low energy generated at the peak load point indicate that the main structure of the GF is damaged.

### 3.7. K-Means Damage Classification

The classification method based on peak frequency and amplitude is a straightforward and efficient approach for categorizing AE signals. This method can accurately distinguish between various damage patterns in composites. Therefore, this paper primarily employs the K-means algorithm to cluster the amplitude and peak frequency of the AE signal. The peak frequency of the AE signal is the frequency at which the signal reaches its maximum amplitude, reflecting the vibration characteristics of the AE source. Various damage modes generate distinct AE signals, resulting in variations in the peak frequency. Therefore, by comparing the magnitude and trend of the peak frequency, it is possible to determine the degree and type of damage to the material. The amplitude magnitude refers to the maximum value of the AE signal’s amplitude. By comparing the amplitude magnitude of different types of damage, it is possible to roughly determine the nature and extent of the damage.

In composite materials, matrix cracking usually results in a lower peak frequency due to the larger strains and stress concentrations associated with this type of damage. Conversely, fiber breakage may result in higher peak frequencies because this type of damage usually entails smaller strains and stress concentrations. Similarly, fiber pullout, fiber relaxation, and delamination damage may have different peak frequency distribution ranges. Specifically, fiber pullout with fiber relaxation may result in intermediate peak frequencies due to interfacial issues between the fibers and the matrix. In contrast, delamination damage may lead to higher peak frequencies as it typically involves defects or defective regions within the material [65,66,67].

Analyzing the K-means clustering results in Figure 12, it is evident that the AE signals from 0–100 kHz in CL1 correspond to the matrix cracking mode, while the high-frequency signals above 600 kHz in CL6 correspond to the fiber breakage mode. The higher amplitude of the 100–250 kHz AE signal in CL2 indicates interface debonding. The 250–380 kHz signal in CL3 and the 380–450 kHz signal in CL4 are classified as mid-frequency signals, positioned between matrix cracking and fiber breakage. The signal distribution does not exhibit a clear boundary, suggesting that the signals within this range are associated with two distinct damage modes: fiber pullout and fiber relaxation, respectively. CL5 contains a high AE signal amplitude of 450–600 kHz, which indicates a matrix delamination damage mode.

### 3.8. Peak-Frequency FFT Transform

K-means cluster analysis focuses on clustering the spatial distribution of signals rather than directly extracting the frequency components of the signals. In contrast, the FFT can convert the signal from the time domain to the frequency domain, providing more intuitive frequency distribution information. Before implementing K-means cluster analysis, it is often necessary to perform preprocessing steps on the data, such as normalization and standardization, which can increase computation time. In contrast, the FFT does not require preprocessing steps and therefore demonstrates greater efficiency when dealing with large amounts of specimen data.

The frequency peak FFT transform of the specimen subjected to bending load is depicted in Figure 13. As depicted in Figure 13a, the time-domain signals exhibit chaotic behavior, making it challenging to discern any identifiable patterns. Therefore, Figure 13b–m show only the frequency-domain signals after the FFT transformation. The obtained peak frequency signals undergo FFT transformation, resulting in the appearance of distinct peak bands when the time domain signals are converted to frequency domain signals using the FFT method. Combined with the results of the appellate K-means clustering analysis, it can be seen that the amplitude of the signal waveform is small when the matrix material is cracked, and the frequency peak is concentrated below 100 kHz, indicating a low frequency. The fiber breakage signal has a high frequency and a wide frequency range, with the peak frequency concentrated at over 600 kHz. The interface debonding exhibits low frequency and high amplitude, with the peak frequency concentrated in the range of 100–250 kHz, and multiple peaks are present. Fiber pullout produces low-frequency, medium-amplitude signals, with frequencies ranging from 250 to 380 kHz. Medium-frequency, medium-amplitude signals are fiber relaxation signals, with peak frequencies ranging from 380 to 450 kHz. High-frequency, medium-amplitude AE signals correspond to matrix delamination damage, with peak frequencies concentrated at 450–600 kHz.

The analysis of Figure 13 indicates that after the GF was modified, the peak frequency range and amplitude signals of the damage extension process were altered. In the unmodified specimen, there are more signals and higher amplitudes in the peak frequency range corresponding to interfacial debonding and fiber pullout, indicating a high degree of damage in this specimen. As the concentration of the NaOH solution increases and the processing time lengthens, the signal within the peak frequency range corresponding to interface debonding and fiber pullout damage gradually decreases or the amplitude decreases. Conversely, the fiber breakage corresponds to an increasing number or amplitude of signals within the peak frequency range. The principle of the reaction between the GF and NaOH can explain this result, which corresponds to the process of damage extension in the modified and unmodified GF/VER composites studied in the previous section. When the concentration of the NaOH solution is increased and the processing time is extended, the interfacial bond strength of the GF/VER composites also increases. As a result, the specimens are subjected to loading conditions that hinder interfacial debonding and fiber pullout, leading to a gradual decrease in the signal within the peak frequency range. When the concentration of the NaOH solution is excessively high and the treatment time is prolonged, it leads to the deterioration of the mechanical properties of GF. Consequently, there is an increase in the damage to fiber breakage in the composites, resulting in a gradual rise in the corresponding signal within the peak frequency range.

## 4. Conclusions

In this paper, the AE technique was used to study the changes in the interface and mechanical properties of GF/VER composites after the GF was treated with NaOH, and the modification effect and damage process of GF/VER composites were characterized by intuitive data and figurative signal characteristics, and the following conclusions can be drawn:(a)The interfacial properties of the GF/VER composites were improved through treatment with a 2 mol/L sodium hydroxide solution for 48 h. Compared with the original samples, the interfacial shear strength of monofilaments increased to 5.00 MPa, while fiber bundles reached 29.04 MPa, representing increases of 25.31% and 27.48%, respectively.(b)After the GF was modified by NaOH solution, the surface of the GF became rough, and the interface between the GF and VER produced physical riveting and chemical bonding, which improved the interfacial bond strength of the material. When the concentration of the NaOH solution reaches 2 mol/L and the treatment time reaches 48 h, the flexural strength of the GF/VER composites reaches a peak value of 346.72 MPa, which is enhanced by 20.96% compared to that before being modified.(c)When the GF/VER composites were subjected to a three-point bending load, the load-time curves strongly correlated with the AE characteristic signals. When the concentration of the NaOH solution reaches 2 mol/L and the treatment time reaches 48 h, the energy and counting signals during the stable growth period decrease, while the energy and counting signals during the damage period significantly increase. Additionally, the energy at the peak loading point also shows a substantial increase. The damage to modified GF/VER composites was improved, resulting in the release of more energy when damage occurred.(d)The peak frequency ranges of the six types of damage processes (matrix cracking, interlaminar debonding, fiber pullout, fiber relaxation, matrix delamination, and fiber breakage) for GF/VER composites subjected to three-point bending loads are 0~100 kHz, 100~250 kHz, 250~380 kHz, 380~450 kHz, 450~600 kHz, and 600 kHz and above, respectively.

## Figures and Tables

**Figure 1 materials-17-01407-f001:**
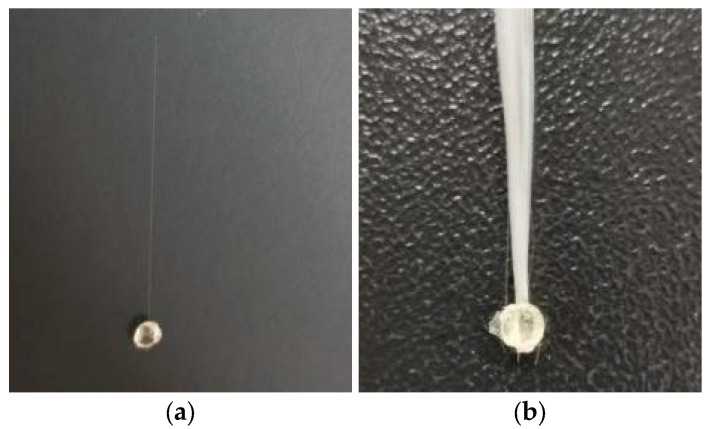
Specimens: (**a**) monofilament pullout specimen; (**b**) fiber bundle pullout specimen.

**Figure 2 materials-17-01407-f002:**
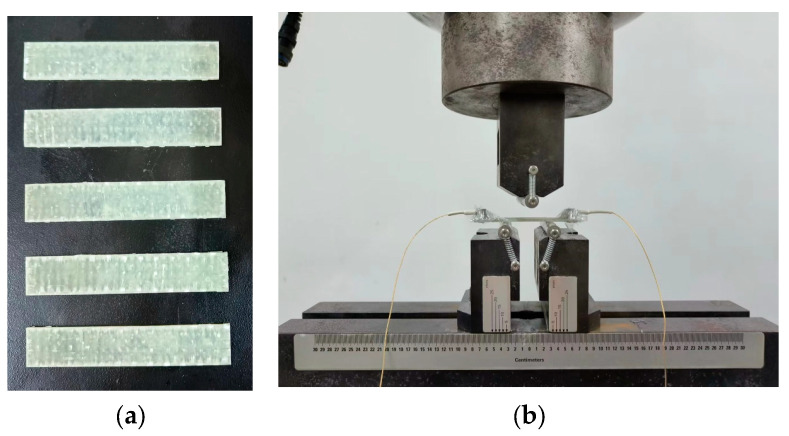
Three-point flexural test diagrams: (**a**) GF/VER composite samples; (**b**) sample under flexural load.

**Figure 3 materials-17-01407-f003:**
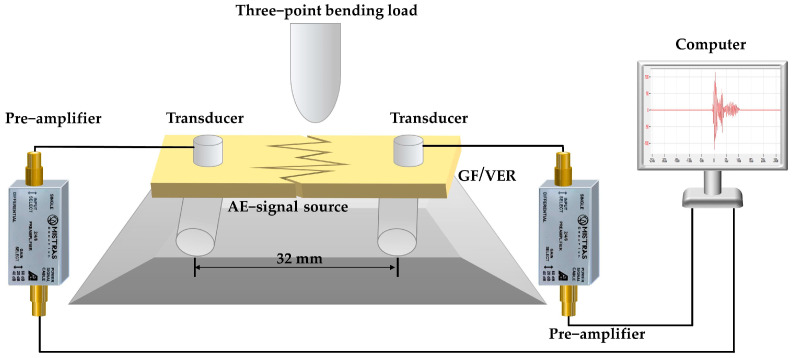
Acoustic emission detection of three-point bending load of GF/VER composites.

**Figure 4 materials-17-01407-f004:**
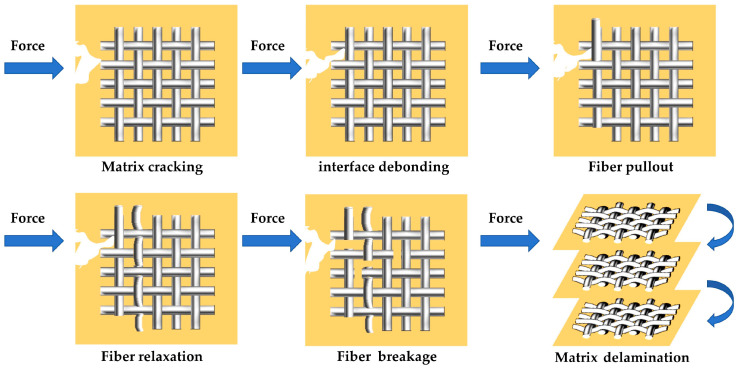
Acoustic emission sources during damage of fiber-reinforced composites.

**Figure 5 materials-17-01407-f005:**
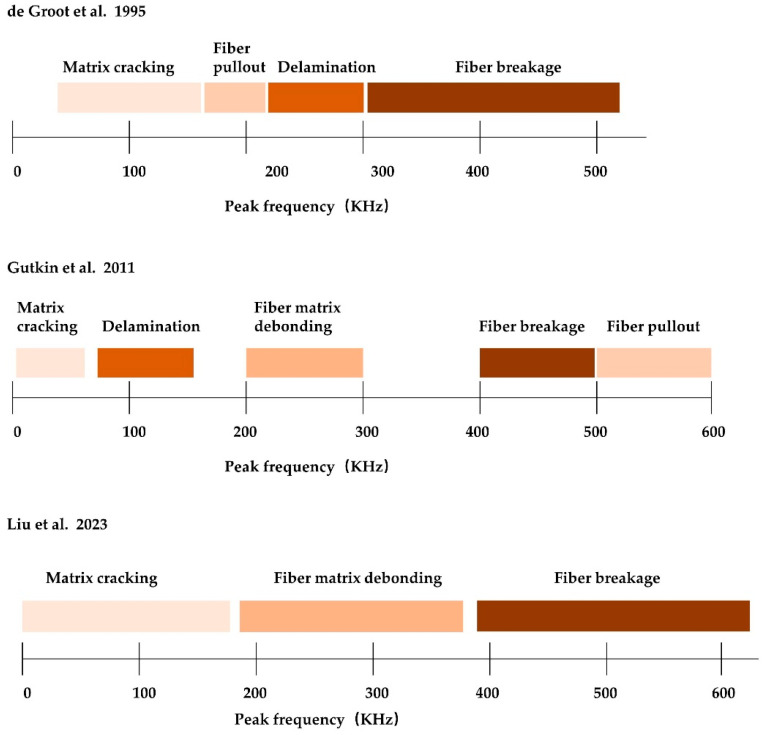
Frequency range definition [34,35,59].

**Figure 6 materials-17-01407-f006:**
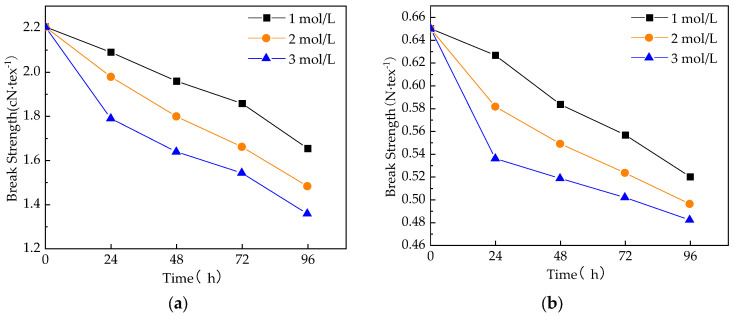
Changes in the breaking strength of unmodified and modified GF: (**a**) monofilament breaking strength, and (**b**) fiber bundle breaking strength.

**Figure 7 materials-17-01407-f007:**
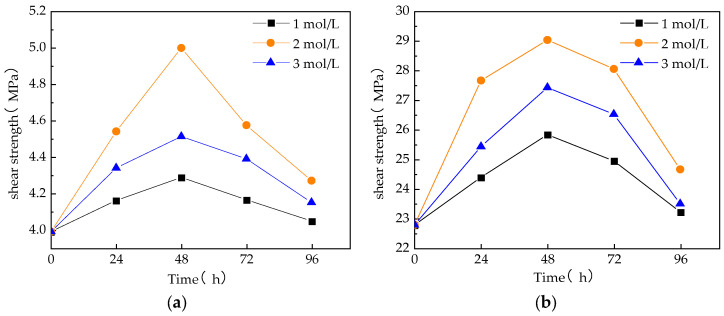
Effect of sodium hydroxide solution treatment on interfacial properties of composites: (**a**) monofilament interfacial shear strength; and (**b**) fiber bundle interfacial shear strength.

**Figure 8 materials-17-01407-f008:**
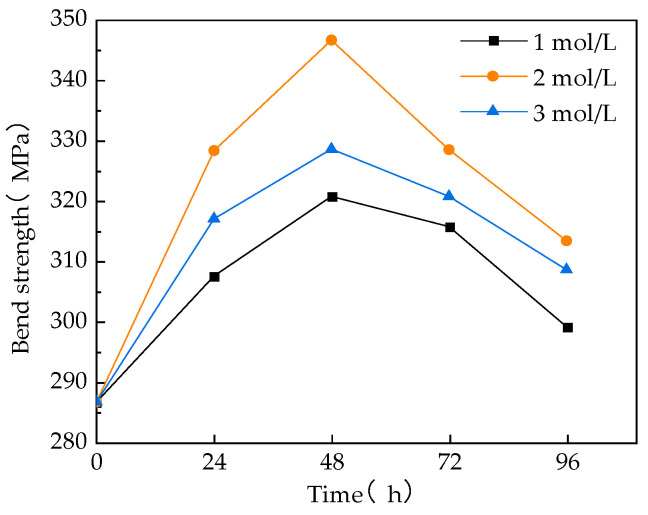
Effect of NaOH solution treatment on flexural strength of GF/VER composites.

**Figure 9 materials-17-01407-f009:**
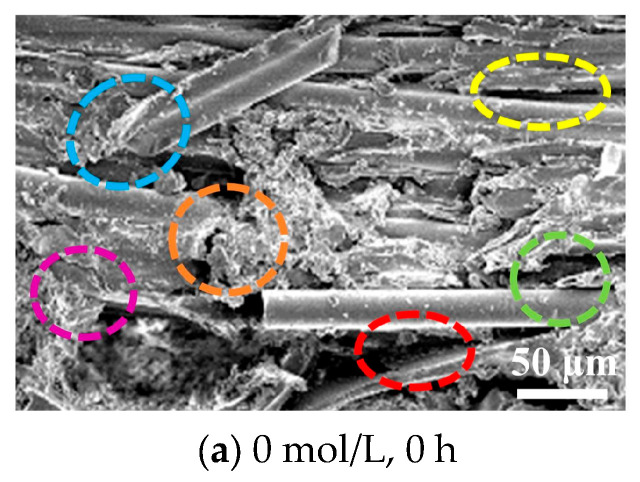
SEM photographs of damaged fracture surfaces of GF/VER composites.

**Figure 10 materials-17-01407-f010:**
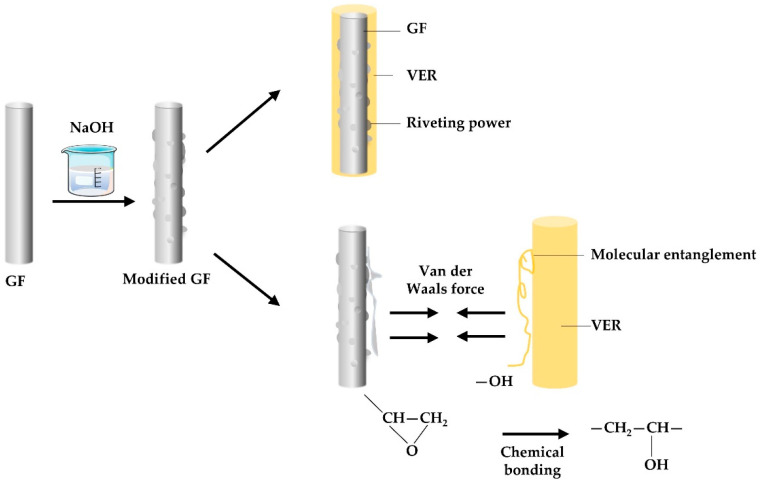
Adhesion between modified glass fiber and vinyl ester resin.

**Figure 11 materials-17-01407-f011:**
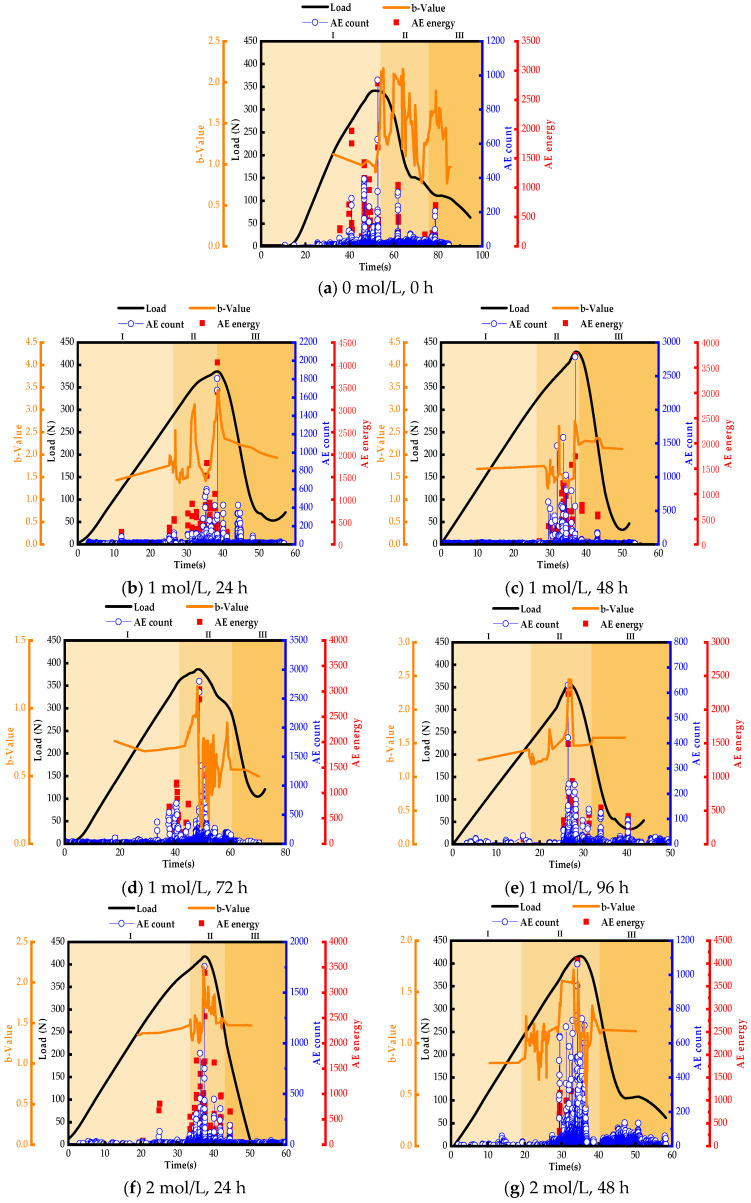
Relationship between load-time curves and AE characteristic parameters.

**Figure 12 materials-17-01407-f012:**
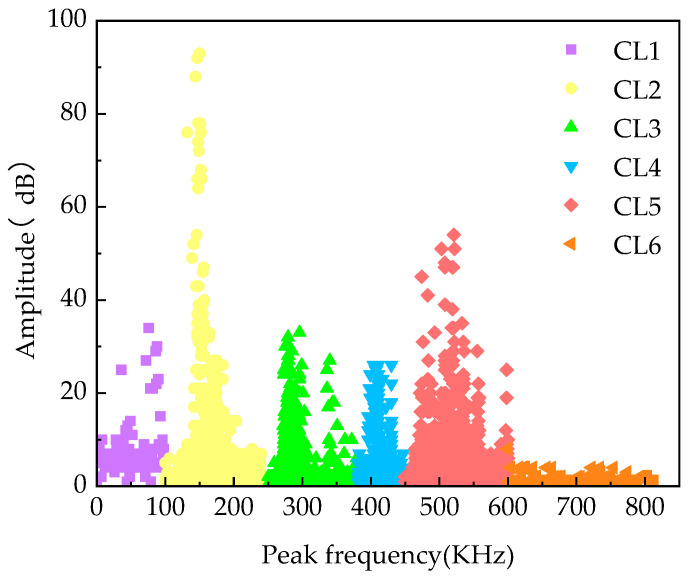
K-means clustering results.

**Figure 13 materials-17-01407-f013:**
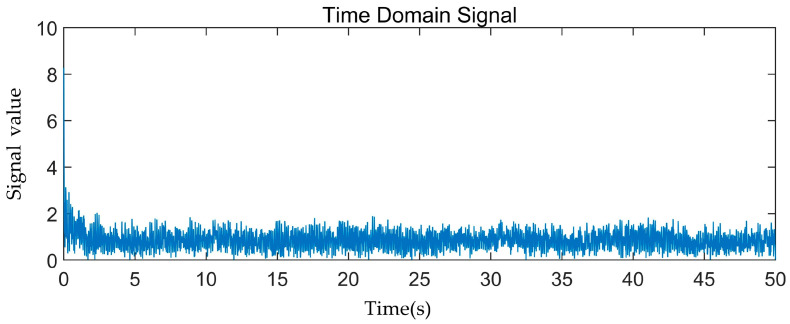
Peak frequency FFT transform.

**Table 1 materials-17-01407-t001:** Materials and equipment.

Name	Parameter Specification or Model	Manufacturer
Glass fiber	Plain weave, thickness of 0.25 mm, warp density of 34 ends/10 cm, weft density of 26 picks/10 cm, Warp and weft yarn density of 450 tex	Jushi Group Co., Ltd. (Jiaxing, China)
Vinyl ester resin	SWANCOR 901, styrene type	Guangzhou Yuexing Enterprise Co., Ltd. (Guangzhou, China)
Acetone solution	The mass fraction is 99.55%	Guangdong Wengjiang Chemical Reagent Co., Ltd. (Guangzhou, China)
Anhydrous ethanol solution.	The mass fraction is 99.5%	Jiangsu Qiangsheng Chemical Function Co., Ltd. (Changshu, China)
NaOH solution	The mass fraction is 95%, granular	Hangzhou Mick Chemical Instrument Co., Ltd. (Hangzhou, China)
Electronic single fiber strength machine	YG001A	Zhejiang Sanshouji Instrument Co. (Wenzhou, China)
Universal material machine	MTS	MTS Industrial Systems (China) Ltd. (Shanghai, China)
Acoustic emission instrument	PAC	American Physical Acoustics Company (Beijing, China)
Scanning electron microscope	JSM-5610LV	Japan Electronics Co., Ltd. (Tokyo, Japan)

**Table 2 materials-17-01407-t002:** AE system parameterization.

Parameter	Setting Value
Threshold/dB	40
Sampling rate/MSPS	1
Peak definition time (PDT)/µs	50
Hit definition time (HDT)/µs	100
Hit locking time (HLT)/µs	300

## Data Availability

Data can be provided upon request from the corresponding author.

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
