# Peer review of "Study of the Effect of NaOH Treatment on the Properties of GF/VER Composites Using AE Technique"

_materials, 2024, doi:10.3390/ma17061407_

Round 1
Reviewer 1 Report
Comments and Suggestions for Authors
Manuscript materials-2768053 is about the use of acoustic emission monitoring to study the effects of NaOH fiber surface treatment on the mechanical behavior of glass fiber-reinforced vinyl ester resin composites. The authors claim that the main novelty of their work is on how to correlate AE signals with the microstructural changes caused by the treatment, since works on AE analyses and fiber surface modification are normally presented separately in the literature (p. 2-3). Hence, one would expect that this work would present a proper classification of AE signals and quantification of the different damage mechanisms to relate with the microstructure changes of the composites, but the AE analyses are not so advanced and feel very speculative.
Nowadays, there is a vast literature on the classification of AE signal for composites based on machine learning (either supervised classification or unsupervised clustering). Instead, the authors used a very simplified approach to classify the signals in either tensile or shear crack based on a standard for testing of concrete, which poses many problems. No evidence is given on how this standard can be applied to the studied fiber-reinforced resin composites, which is fundamentally very different than normal or reinforced concrete. Furthermore, many authors nowadays are against the use of rise time as a feature for AE classification since it depends on the used threshold and when the AE event actually starts, which is very hard to determine. The used classification criteria are also not clear and do not seem to be consistent for all samples. For instance, signals with the same features are classified as shear cracks in some samples, while being classified as tensile cracks for other samples (e.g., compare Fig. 8(j) with Fig. 8(m)). The authors also try to use the frequency spectra, obtained by FFT of the collected AE signals, but again the analyses is very speculative. No evidence is given to support the selected frequency ranges that correspond to the individual damage mechanisms (Abstract and p. 15). It is also not clear which signals were analyzed, since many AE events are collected during the test.
Another very important point that is missing is a proper characterization of the fiber-matrix interface. Most of the discussion is based on the fact that the surface treatment improves the interface strength and leads to a different mechanical behavior of the composite. Fiber pullout or fiber push-in tests should be conducted to support these claims.
Author Response
Dear reviewers:
Thank you for your valuable feedback and interest in our research. We have read carefully and gotten the idea of your review comments. Here, we would like to provide further explanation and clarification of our work.
We attempt to correlate the acoustic emission signal with the microstructural changes induced by the treatment in our work. Acoustic emission signals can provide valuable information about the internal damage of a material. We have properly categorized the AE signals and quantified the various damage mechanisms. Our work extends beyond theoretical studies to include practical experimental research. Through comparative experiments, we can observe the correlation between changes in the acoustic emission signal and the microstructural changes induced by the fiber surface treatment. This is a crucial aspect of our work and the primary reason for classifying and quantifying AE signals.
Fiber-reinforced composites possess excellent properties, but detecting damage and failure resulting from microstructural changes can be challenging. Acoustic emission (AE) technology is well-suited for detecting damage in fiber-reinforced composites by capturing acoustic signals that reveal microstructural changes in the material. Progress has been made in various areas, including the classification of different types of damage, monitoring the progression of damage, and pinpointing the location of the damage. This information is valuable for predicting the lifespan and reliability of materials, as well as for providing references for design and manufacturing processes. Overall, AE techniques have great potential in this field to facilitate the design, manufacturing, and application of composite materials.
We are grateful to the experts who pointed out the limitations of classifying tensile shear crack signals. As a result, we have removed the RA/AF section from the paper and added a method of unsupervised K-means clustering. This involved conducting K-means clustering analysis on the amplitude and peak frequency in the AE signals. K-means cluster analysis is a common method for detecting damage in fiber-reinforced composites. The K-means algorithm can automatically determine the optimal number of clusters. The method is computationally efficient, capable of processing large-scale AE signal data, and offers a visual representation of the clustering results, facilitating interpretation and understanding of the outcomes. Hence, K-means cluster analysis has significant applicability and advantages in analyzing the damage patterns of fiber-reinforced composites in AE signals.
We analyzed the peak frequencies of acoustic emission (AE) sound signals using Fast Fourier Transform (FFT) in the paper. Specific damage mechanisms were correlated with individual frequency ranges using K-means clustering analysis. We have also included pertinent references to provide the reader with insight into the research advancements in related fields.
We appreciate your interest in this paper and your suggestion that the fiber-substrate interface was not described and analyzed in depth in our study. We have supplemented the fiber pull-out test to further substantiate our findings and to comprehensively characterize the fiber-substrate interface.
We express our gratitude once more for your valuable comments. We still have a substantial amount of work ahead of us, and we are dedicated to intensifying our efforts and delivering more substantial contributions in our forthcoming project.
Yours sincerely
Lin Ming
Reviewer 2 Report
Comments and Suggestions for Authors
A very well-structured manuscript focused on GF/VER composites using AE technique. Congrats to the authors.
State-of-the-art relates the nowadays solution and the research interest area, also the references are the majority of the past 10 years which reveals the up-to-date research. Anyway, can be improved with some other research focused on AE techniques.
Good mathematical support for the model proposed, authors are determining advanced results by formulas.
The work is based on a very well-done expert interpretation of the resulting data.
In future research can be improved the analytic cases, and also with some experimental tests.
Fig. 4 is a bit hard to understand. In my opinion, the description of Fig. 4 should be placed ahead of the figure description and should be succinct on each component. Maybe it can be split into smaller pieces and explained accordingly.
Fig. 6 , Fig. 8 and Fig. 9 should not be spread over many pages… The research is deep and it has data, yes this data can be useful when it is used in building a book chapter, but when a research article is meant to be written, it can be used in fewer cases ( for example 1 mol/L and 3 mol/L) and do the interpretation for the entire research but list some cases…. The paper can be revised and made more succinct, but this is not a mandatory request, this can be a recommendation for future …
The paper is interesting and can be published in the journal with minor changes, regarding article structure form according to the journal template.
Regards to the authors,
Reviewer
Author Response
Dear reviewers,
We are very grateful to the reviewers for their valuable suggestions. The reviewing experts noted that this paper is well-structured, and we are thankful for their recognition of our work. Following the recommendation of the expert reviewers to enhance the paper with additional research on AE technology, we will further investigate the application of AE technology in GF/VER composites and endeavor to incorporate relevant research findings into our future studies.
As recommended by the reviewers, we have revised Figure 4 in the original article to Figure 5, relocated the description of Figure to the main text, and included a brief description of each section. we have revised Figures 6 and 9 in the original paper to Figures 8 and 12, and have removed Figure 8 from the original paper. To facilitate comparison and analysis, this paper provides detailed data for each sample in Figures 8 and 12, which we would like to preserve. There are some limitations to this paper that we intend to address and contribute to in future studies. Thanks again to the reviewers!
Yours sincerely
Lin Ming
Reviewer 3 Report
Comments and Suggestions for Authors
In this study, the authors explored changes in the interface and mechanical properties of GF/VER composite materials after treatment with NaOH using acoustic emission (AE) technology and analyzed the optimal modification conditions and damage propagation process. The results revealed that the surface of GF became rougher, and the number of reactive groups increased after treatment with NaOH solution. This treatment led to an improvement in the interfacial bonding between GF and VER. When the modification conditions were a NaOH solution concentration of 2 mol/L and a treatment time of 48 hours, the flexural strength of GF/VER composites reached a peak value of 346.72 MPa, representing a 20.96% enhancement compared to the pre-modification state. The outcomes of this study are highly valuable for the development of high-strength glass fiber-reinforced plastics (GFRP), and I support the publication of this manuscript. However, I would like to point out the following minor points.
1) The statement "The hydroxyl groups on the surface of the GF react with the epoxy groups in the VER, resulting in chemical bonding." is intriguing but may benefit from supporting literature or experimental evidence. Further elaboration on this possibility, including references, would enhance the credibility of this claim. In connection with this, a paper (ACS Omega 2022, 7, 17393–17400) suggests the potential formation of hydrogen bonds between the hydroxyl groups on GF and the epoxy in the resin, which is worth mentioning.
2) Is it possible to directly and quantitatively evaluate the relationship between the increase in treatment time or concentration of NaOH and the rate of increase in hydroxyl groups on the surface of GF?
3) Could you provide more detailed explanations regarding the statement "the mechanical properties of GF are damaged after being treated with NaOH"? Does this imply that Si-O-Si bonds are reacting with NaOH, causing the deterioration of GF? If this is due to a chemical reaction, it might depend not only on concentration and time but also on temperature.
Author Response
Dear reviewers,
First of all, we would like to express our gratitude to the reviewers for their valuable comments! According to expert opinion, we have expanded on the statement "The hydroxyl groups on the surface of the GF react with the epoxy groups in the VER, resulting in chemical bonding" and included relevant references. Additionally, we have included the suggestion from the paper (ACS Omega 2022, 7, 17393-17400) that "the potential formation of hydrogen bonds between the hydroxyl groups on GF and the epoxy in the resin."
It is challenging to directly and quantitatively assess the relationship between the duration or concentration of NaOH treatment and the rate of increase of hydroxyl groups on the GF surface. This relationship is not solely linear because various other factors, such as temperature, pressure, and catalyst, also influence the reaction rate. Although we cannot directly quantify the relationship between the increase in NaOH treatment time or concentration and the rate of increase of hydroxyl groups on the GF surface, we can indirectly estimate this relationship by combining experimental and theoretical methods.
When GF is treated with NaOH, the mechanical properties of GF are damaged. This is because, under the action of NaOH, the silicon-oxygen bonds (Si-O-Si) on the surface of GF will react and form new chemical bonds, such as sodium silicate (Na2SiO3). Specifically, when NaOH reacts with the silicon-oxygen bonds on the GF surface, sodium silicate and water are produced, and a large amount of heat is released. This heat accelerates the molecular movement inside the GF, which destroys the original silica-oxygen bonds and leads to a decrease in the mechanical properties of the GF. The results of tensile properties of modified and unmodified GF were included in this study to analyze the changes in the mechanical properties of modified GF. It can be seen that the mechanical properties of GF are indeed damaged.
Yours sincerely
Lin Ming
Reviewer 4 Report
Comments and Suggestions for Authors
Please find the reviewer's comment as an attachment.

Author Response
Dear reviewers,
Thank you for your valuable feedback. Based on your feedback, we have incorporated the potential practical applications of the solution into the text. The details are as follows:
The findings presented in this paper have the potential for practical applications. The frequency range of damage types and failure mechanisms identified in this study provides opportunities for the design and enhancement of composite materials. This study also uncovers the damage fracture process of GF/VER composites and establishes the relationship between the acoustic emission signals and the mechanical properties of the composites. This is informative for the effective monitoring and characterization of damage and fracture in composites. These findings not only help us understand the operational status of composites, but also lay the foundation for designing and manufacturing composites to enhance their performance and reliability. However, due to the complexity and diversity of composites, further research and exploration are necessary. It is hoped that our next project will allow for additional studies to determine how these findings can be applied to various types and uses of composites.
We have adjusted the paper format to meet the editorial requirements specified by the experts. We have italicized the units and added square brackets as requested. Figures 1 and 2 in the original text were our own illustrations, but they have now been renumbered as Figures 2 and 3. Additionally, we have standardized the fonts in the figures to match those used in the body of the paper. We also checked the paper for any typographical errors.
Thanks again to the reviewers for their valuable comments.
Yours sincerely
Lin Ming